# COVID-19 Vaccination Acceptance and Hesitancy in Healthcare Workers and the General Population: A Systematic Review and Policy Recommendations

**DOI:** 10.3390/ijerph21091134

**Published:** 2024-08-28

**Authors:** Alessandra Pereira da Silva, Luciana Ribeiro Castaneda, Ana Paula Cavalcante de Oliveira, Inês Fronteira, Isabel Craveiro, Leila Senna Maia, Raphael Chança, Mathieu Boniol, Paulo Ferrinho, Mario Roberto Dal Poz

**Affiliations:** 1Instituto Nacional do Câncer, Rua Marquês de Pombal 125, 12° andar Centro, Rio de Janeiro 20230-240, RJ, Brazil; raphael.chanca@inca.gov.br; 2Instituto de Medicina Social, Universidade do Estado do Rio de Janeiro, Rua São Francisco Xavier, 524-7° andar, Blocos D e E–Maracanã, Rio de Janeiro 20550-013, RJ, Brazil; luciana.ribeiro@ifrj.edu.br (L.R.C.); apco.hrh@gmail.com (A.P.C.d.O.); senna.maia@gmail.com (L.S.M.); dalpozm@uerj.br (M.R.D.P.); 3Global Health and Tropical Medicine, Instituto de Higiene e Medicina Tropical, Universidade NOVA de Lisboa, Rua da Junqueira 100, 1349-008 Lisboa, Portugal; ines.fronteira@ensp.unl.pt (I.F.); isabelc@ihmt.unl.pt (I.C.); pferrinho@ihmt.unl.pt (P.F.); 4Health Workforce Department, World Health Organization, Av. Appia 20, 1202 Geneva, Switzerland; boniolm@who.int

**Keywords:** COVID-19, vaccination coverage, vaccine hesitancy, vaccination refusal, healthcare workers

## Abstract

Introduction: The COVID-19 pandemic challenged the scientific community to find and develop a vaccine to fight the disease. However, problems with achieving high vaccine coverage have emerged, even among high-risk groups such as healthcare workers (HCWs). Objective: The objective of this study is to examine factors that influence HCW’s and the general population’s adherence to COVID-19 vaccination and national policies to vaccinate HCWs and other target groups. Methods: This study implemented a systematic review. The eligibility criterion for inclusion was being a HCW, target population for COVID-19 vaccination, or general population. Vaccination was the target intervention, and the COVID-19 pandemic was the context. We selected publications published between 1 January 2020 and 31 March 2022. Qualitative synthesis used a meta-aggregation approach. Results: Nineteen articles were included in the review, with study samples varying from 48 to 5708 participants. Most of the evidence came from cross-sectional and qualitative studies. The main findings were related to vaccine hesitancy rather than acceptance. Factors associated with HCW vaccine hesitancy included subjective feelings such as safety concerns, rapid vaccine development, and insufficient testing. Countries have adopted few public policies to address this problem, and the main concern is whether to enforce vaccination and the extent to which measures are legal. Conclusion: The quality of the evidence base remains weak. Skepticism, mistrust, and hesitancy toward vaccination are global issues that can jeopardize vaccination coverage.

## 1. Introduction

The COVID-19 pandemic was an unprecedented public health emergency that continues to challenge the scientific community to develop mitigating strategies, including vaccines, to overcome the emergency. The COVID-19 vaccine deployment has revealed the additional need to develop adequate means to achieve vaccine coverage. Low-income countries (LICs) were unable to obtain sufficient vaccines to immunize their entire populations. Despite increased access to vaccines, high- and middle-income countries (MICs) have faced policy indecisiveness in managing their healthcare systems [1].

The global consequences of the COVID-19 pandemic are still challenging countries. The World Health Organization (WHO) reported a total of 628,346,704 cases and 6,573,968 deaths from the disease from 2019 to 3 November 2022. The United States of America (USA), Brazil, India, Russia, and Mexico have the highest absolute numbers of deaths [2]. Healthcare systems have thus sought various alternatives to control the pandemic.

Healthcare workers are workers in public and private health services at different levels of complexity. Thus, it includes both health professionals, support workers, and others—that is, those who work in health services but who are not providing direct healthcare services to people. It also includes those professionals who work in home care, such as caregivers for the elderly and doulas/midwives.

The general population comprises people from all countries, without demographic cuts. The selection of target groups in accordance with the vaccination strategy and policy recommendations is of crucial importance. For example, groups at the highest risk of death should have priority for vaccination. Likewise, rapid diagnostic tests and vaccines must be intended as a priority for providers of essential services in a scenario where the preservation of these services is essential.

The most promising approach was to initially provide for the development of immunity in target groups and the general population through vaccination strategies to reduce the number of seriously infected people and case-fatality, subsequently scaling up the strategy to achieve population immunity. Between 2020 and 2021 in LICs, due to the lack of vaccines, the threats associated with COVID-19 and the various SARS-CoV-2 variants were a concern. Thus, in addition to other measures to prevent infection with the novel coronavirus, it is crucial to vaccinate both the general population and healthcare workers (HCWs) that perform vaccination and provide other healthcare services.

High-income countries secured their vaccine stocks in advance by pre-purchasing millions of doses of vaccines even before the clinical trials were finalized. Asian countries such as Japan and Australia have also followed this trend and secured their vaccine doses. However, for low-income countries, such as those in Africa, the trend of purchasing and guaranteeing early access has not occurred. As a result, more than 3 billion doses of the COVID-19 vaccination had been administered worldwide by July 2021, with only 0.9% of people in low-income countries having received at least one dose [3].

Healthy HCWs are essential for health systems’ resilience and effectiveness, as HCW actions determine the efficient use of other resources. The HCW is thus central to the elaboration of health policies, including for COVID-19. The way countries deal with the COVID-19 crisis depends greatly on the effective protection and deployment of the HCWs. Many measures can and must be taken to prepare the HCWs to defeat the pandemic [4,5].

HCWs are at high risk of occupational SARS-CoV-2 exposure and transmission, which prioritizes them for early COVID-19 vaccination. However, there has been an alarmingly high rate of COVID-19 vaccine hesitancy in the HCWs [6,7]. Recent evidence suggests that refusal to be vaccinated is associated with concerns about efficacy, safety, and the unprecedented fast-track vaccine production process [6,7]. Since HCWs serve as role models of appropriate preventive health behavior for the public at large, the refusal by significant numbers of HCWs to be vaccinated against COVID-19 may also signal danger for the population [8] and contribute to widespread vaccine hesitancy. Vaccine hesitancy is a significant barrier to vaccine uptake and the achievement of collective immunity via vaccination.

Despite available data on the safety and effectiveness of the COVID-19 vaccines, skepticism, mistrust, and hesitancy are global problems. The unvaccinated portion of the population can prevent achieving collective immunity via vaccination [9]. As discussed in this article, the overall impact on society and the factors that can explain non-vaccination have been explored in the literature in the last two years. Numerous factors can lead to low vaccination coverage in the general population, such as weak physician–patient communication, gaps in COVID-19 vaccine knowledge, perceived susceptibility, and the perceived severity of the vaccine’s side effects [10]. The lack of previous experience with a global vaccination campaign of this magnitude created a knowledge gap that still needs to be addressed to inform health policies and multisector decisions that can affect vaccine uptake worldwide.

The study aims to examine factors that influence HCWs’ and the general population’s adherence to vaccination and elucidate policies and strategies to cover health personnel and the population at large with the COVID-19 vaccination.

The focus of the research was on health professionals, the general population, and vaccination. The scope was expanded to facilitate search methods and find as many articles as possible. This systematic review is based on four questions (Box 1).

Box 1Review questions. What are countries’ requirements to ensure COVID-19
vaccination coverage for the target population? How are countries managing COVID-19 policy, regulation,
prioritization, and mandatory vaccination of the HCW and the target
population? What are the barriers and mechanisms involved in
improving vaccination teams’ performance and productivity? What are the main barriers and enablers for countries to
obtain the desired or required COVID-19 vaccination coverage for HCW and the
target population?

The four questions are complementary and aim to analyze the problem comprehensively, i.e., workers’ skills to vaccinate the population, how to qualify teams to vaccinate and increase productivity, obstacles and facilities for health workers to accept vaccination, and strategies and policies used to vaccinate health workers and the general population.

## 2. Methods

Research planning was guided by the PRISMA 2020 statement, and the protocol of the present study was registered in the PROSPERO database (https://www.crd.york.ac.uk/prospero/display_record.php?RecordID=325111, accessed on 2 May 2024).

The PICo search tool (Population, Intervention/phenomenon of interest, Context) was used for the questions and data extraction to perform this systematic review [11].

### 2.1. Eligibility Criteria

#### 2.1.1. Population

The review included all studies in which participants were general or target populations or healthcare workers. Healthcare workers include both health professionals, support workers, caregivers, and doulas/midwives. The general population comprises people from all countries, without demographic cuts. The selection of target groups in accordance with the vaccination strategy and policy recommendations is of crucial importance. Due to the highest risk of severe symptoms or death, the target population should have priority for vaccination and be included in the population.

#### 2.1.2. Intervention/Phenomenon of Interest

The systematic review focused on research policies, barriers, and enablers for COVID-19 vaccination coverage, performance, and productivity.

#### 2.1.3. Context

The context was the COVID-19 pandemic and vaccination of the HCWs and the general or target population.

The inclusion criteria follow the PICo guidelines (Appendix A). The bibliographic and gray literature included databases, articles, and internet pages. The selected studies included systematic reviews, quantitative observational studies (i.e., cohort, case-control, and cross-sectional), mixed methods, and qualitative studies published in English, French, Hindi, Portuguese, Italian, or Spanish. The target outcomes comprised policies, strategies, and interventions adopted in countries and mechanisms to improve vaccination coverage among HCWs, as well as strategies to achieve vaccination coverage for the respective population groups. We selected articles published between 1 January 2020 and 31 March 2022.

The following were considered eligible for research: studies in which the target population is or includes HCWs; studies that address vaccination against COVID-19; studies that address the requirements of health professionals for the population to accept the vaccine; studies that describe policies and strategies for vaccine acceptance by the population; studies that describe approaches to improving the quality and productivity of vaccination teams; and studies that describe facilitators and barriers to COVID-19 vaccination among healthcare workers. Figure 1 shows the eligibility flowchart for the adopted criteria.

The eligibility criteria were first applied to the title and abstract/executive summary or introduction of the studies and, if met, applied again to the full texts (Appendix A). The full text was assessed for eligibility in case no abstract or equivalent was available.

### 2.2. Information Sources and Search Strategy

The following databases were searched for this review: MEDLINE-PubMed, Embase, Scopus, Latin American and Caribbean Health Sciences Literature, and Web of Science. The World Health Organization Database and Google Scholar were adopted for the gray literature. The controlled vocabularies for the health area DeCs (Descriptors in Health Sciences), MeSH (Medical Subject Headings), and Emtree (Embase Subject Headings) were consulted to retrieve findings. The search terms used were a combination of official descriptors and free terms, extracted from the controlled vocabularies Mesh, DeCS, and Emtree, which enabled the construction of search strategies for each database adopted. Table 1 illustrates the terms used to construct the search strategy in the PubMed database.

Each review question had a specific search strategy per database and was developed by a librarian with expertise in systematic reviews. Appendix A provides the entire search strategy.

### 2.3. Selection of Studies

Endnote was used in the selection process to collect, organize, and manage references retrieved by the searches in the various databases [12]. Once this phase was complete, we uploaded the reference file to Rayyan [13]. Two researchers (APS and LC) performed the selection independently, first by reading titles and abstracts and then by evaluating the full texts using the Rayyan program. Differences were discussed with a third researcher until a consensus was reached.

Eligibility and exclusion criteria were first applied to the abstracts of the retrieved documents, independently by at least two reviewers. When no abstract was available, the criteria were applied to the executive summary or the article’s introduction.

When the eligibility criteria were met, documents were selected for full-text analysis. In this phase, the eligibility and exclusion criteria were applied again, and if they were met, relevant data were then extracted from the document. In cases of disagreement, a discussion between the two reviewers was conducted, and a third reviewer was consulted. Kappa statistics were computed to assess inter-reviewer agreement and the quality of this process. Kappa inter-reviewer agreement was 9.4. Kappa values were calculated to measure agreement between two reviewers (APS and LC). The reviewers considered the standard error, level of agreement, and respective 95% confidence intervals (source of instrument: www.vassarstats.net; (accessed on 2 May 2024)) [14].

The inclusion of studies followed the PRISMA guidelines [15], which include the number of documents retrieved, the number of duplicates, the number of documents screened for abstracts or the equivalent, the number of documents included for full-text analysis, the number of excluded documents during full-text analysis, and the number of documents for data extraction.

### 2.4. Data Extraction

Data extraction from the included studies was randomly distributed between two reviewers and performed using a spreadsheet in the REDCap software v12.2.11, as shown in the Appendix A [16]. Appendix A show the information extracted from the selected articles, including author, year, country, title, language, journal, study design, and sample. There was a particular concern about identifying gender-specific aspects referred to in the data.

### 2.5. Assessment of Bias Risk

To assess the risk of bias in the included studies, we used the GRADE CERQual tool (Confidence in Evidence from Reviews of Qualitative research) for qualitative studies [17] presented in Appendix A. The reviewers used an electronic form of the GRADE-CERQual, developed in RedCap and specific to the study design (Appendix A).

### 2.6. Data Synthesis

Qualitative synthesis using a meta-aggregation approach for each research question, organized within each question per outcome and each outcome per type of study.

## 3. Results

We identified 29 texts, all in English, that met the inclusion and quality criteria for data extraction. After the two reviewers critically appraised the 29 full texts, ten were excluded from the assessment of findings and degree of confidence (one failed to meet the study design criteria (opinion report), four were news articles, and five were not clear about the methods). The research’s PRISMA flowchart is shown in Figure 2.

The number of articles identified from the databases, disagreement between the two researchers on the sensitivity test, and reasons for the exclusion of articles per review question are shown in Appendix A. Some studies addressed more than one review question, so absolute numbers are listed according to the reasons for each question. The reasons for the exclusion of the articles are in Appendix A. After full text reading, twelve of the included studies were cross-sectional surveys, five were qualitative, and two were systematic reviews. Years of publication were 2021 (n = 10) and 2022 (n = 9). Some publications answered more than one review question, and there were few studies about policies, regulations, and mechanisms to improve vaccination teams’ productivity and performance.

The studies covered 14 countries, i.e., the USA (7), Italy (2), Turkey (2), Bahrain (1), Hong Kong (1), Israel (1), India (1), Korea (1), Mongolia (1), and the United Kingdom (UK) (1). There were no low- or middle-income countries (MICs) or any from the Southern Hemisphere. There was only one multinational study with “Arabian countries”, “European countries”, and “other countries”.

Appendix A shows the data extraction from the included studies about the general or target population, while Appendix A shows the studies on health care workers.

### 3.1. Health Workers’ Behavior towards Vaccination

Most of the articles that aimed to identify HCWs’ skills for increasing vaccination rates were based on the search for factors associated with vaccine hesitancy. Thus, most reports suggest behaviors that help achieve vaccination coverage targets. For health systems, failure to increase vaccine uptake may be a tragic public health legacy of the COVID-19 pandemic [18]. Low vaccination adherence might be influenced by the behaviors of family members, close friends, and healthcare workers, according to a survey that looked at non-vaccination. These habits can include relying too much on social media, newspapers, and television and mistrusting appropriate vaccination storage [8]. According to a US survey, knowledge of vaccines had a positive and indirect impact on vaccination intention, lowering hesitation to vaccinate, even though anticipated sensitivity to adverse effects was a negative predictor of intention to vaccinate against COVID-19 [19].

Communication is an important skill. Physicians can initiate conversations on COVID-19 vaccination with their patients either at the in-person point of care or via online communication platforms [19]. Vaccine hesitancy can be addressed by paying careful attention to the application of vaccination programs, the correct and effective use of social media, the transparent and precise management of policy processes, and the provision of evidence-based information on vaccines [20].

Al-Metwali et al. (2021) reported that although vaccine hesitancy remained prevalent, the healthcare workforce showed positive attitudes towards the upcoming COVID-19 vaccines. Positive predictive factors included male gender, age, and employment in the medical field. Conversely, vaccine hesitancy was more prevalent among women and nurses. Vaccine acceptance was facilitated by prior influenza vaccination history and self-perceived risk. Additionally, mistrust of the government and worries about the efficacy, safety, and effectiveness of vaccines were obstacles [10].

Appendix A shows the details of the CERQual GRADE evaluation, listing the results of the study on the public or target population, and Appendix A presents the details of the CERQual GRADE evaluation, listing the findings of studies on healthcare professionals.

### 3.2. Policy, Management, and Regulation for Vaccination

The searches in this review identified a few studies on national policies, regulations, and management of the pandemic. Most of the studies discussed mandatory COVID-19 vaccination (and the legality of this measure) and vaccine acceptance by HCWs. Most literature preferred maintaining high coverage without making vaccination mandatory. Thus, efforts to promote vaccine confidence and proactive vaccine uptake are still needed [21].

It is important to keep in mind that, in a study conducted in Mongolia, the country’s agreement rate regarding required occupational immunization was greater than that of other nations [22]. The authors found that healthcare professionals in Mongolia had a strong desire to get the COVID-19 vaccination, which was correlated with their assessment of risk and motivation for protection as well as their level of faith in the vaccine’s efficacy.

### 3.3. Barriers and Mechanisms to Improve Vaccination Teams’ Performance and Productivity

Carpenter D.M. et al. (2022) conducted a cross-sectional survey during the initial distribution of the COVID-19 vaccine in the southern United States and found that the majority of rural pharmacies were interested in and prepared to administer the doses. However, this survey found that many rural pharmacists reported some reluctance about getting the vaccine themselves, and few had received training on COVID-19. Depending on the pharmacy’s and pharmacist’s specific qualities, the number of vaccines they could deliver varied. The number of vaccines that pharmacists could administer varied according to the pharmacy’s and pharmacist’s characteristics [23].

In a hospital in the Republic of Korea, Kim M.H. et al. (2021) assessed the viewpoints of medical experts regarding the ChAdOx1 nCov-19 vaccine. When asked when they made the decision to receive the second dose, 57.1% of the participating HCWs stated that a hospital-wide campaign motivated them. The authors claim that a customized intervention plan based on a survey can enhance COVID-19 immunization adherence [24].

In a different study, Riccò M. et al. (2021) aimed to examine how Italian occupational physicians accepted the SARS-CoV-2 vaccine in terms of their individual knowledge, attitudes, and practices. The authors showed that vaccinations were widely accepted, and most physicians supported making vaccinations for HCWs mandatory. Based on these findings, the authors hypothesized that increasing vaccination rates in work environments could be beneficial [9].

In order to evaluate healthcare professionals’ attitudes towards COVID-19 vaccination and investigate factors associated with their concerns regarding the safety, efficacy, and effectiveness of the vaccine, Li M. et al. (2021) conducted a rapid systematic review. The authors concluded that, to boost healthcare workers’ rates of COVID-19 vaccination uptake, tailored communication tactics are required. They also stress the need for more transparent data and information regarding the efficacy and safety of vaccines [25].

### 3.4. Barriers and Enablers of HCW Vaccination Coverage

A qualitative study by Harrison J. et al. (2021) examined the causes of COVID-19 vaccine reluctance among American nursing facility employees in order to identify potential mitigating factors. The authors state that among nurses, the primary causes of low vaccination rates were mistrust of the government in general, concerns about pre-existing medical disorders, and the perception that the vaccine had been developed too hastily and without conducting enough testing [26].

A survey was conducted in several Turkish provinces by Aci O.S. et al. (2022) to examine in depth the attitudes of healthcare professionals regarding the COVID-19 vaccination. The authors disclosed that certain health professionals experienced negative feelings because of vaccination. HCWs’ views on COVID-19 vaccines have been affected by the negative emotions and burnout they have experienced during the pandemic. The vaccination process was impacted by contraindications and uncertainties regarding the duration of protection. The balance between the benefits and harms of vaccines, as well as myths and prejudices about vaccines, were also significant concerns. Healthcare professionals expressed the need to establish confidence in vaccination. They suggested that different vaccine options should be offered, that the scheduling and notification system for vaccination should be improved, that evidence-based information about vaccines should be provided, and that a safe environment should be created [20].

An extensive survey carried out by Qunaibi et al. (2021) aimed at Arabic-speaking HCWs living in different countries around the world revealed high vaccine hesitancy rates. The main reasons were concerns about the accelerated production and safety of vaccines and distrust in both published studies and health policies [6]. This hesitancy could impede efforts to achieve widespread vaccination and herd immunity.

Another study involving hospital staff carried out in the Republic of Korea found that HCWs’ hesitancy to receive a second dose was significantly associated with being under 30 years of age and that concerns about the vaccine were less common among those who trusted the vaccine’s efficacy and safety. Among those who received the first dose, 96.2% completed vaccination with the second dose [24].

The study by Toth-Manikowski et al. (2022) demonstrated that the vaccination decisions of health professionals were more influenced by the opinions of their colleagues and other people close to them than by the mass media, which highlights the importance of health institutions working internally to promote relationships and build trust among HCWs across departments and positions, especially among doctors and nurses who remain highly trusted in their communities [27].

In a survey conducted at a hospital in Bahrain with primary care doctors and nurses, the authors concluded that healthcare professionals should be encouraged to get vaccinated themselves and to initiate discussions with patients about COVID-19 vaccines. Vaccine safety data in patients with chronic diseases, training, and guidance for junior doctors can make it easier for family physicians to recommend COVID-19 vaccination [28].

### 3.5. General and Target Populations’ Attitudes Regarding Immunization

Vaccinating the population to contain the pandemic was a priority that should be possible for all countries. Various obstacles have hindered vaccination teams’ performance. Knowing the reasons for such obstacles may help solve the problem.

Abou L.R. et al. (2021) described a successful planning experience with immunization in a primary care environment. Their study revealed that patients who were hesitant to receive the COVID-19 vaccine were subjected to an organizational intervention that included reassurances from the doctors about the vaccine and training on how to communicate effectively with patients. The authors revealed that following the intervention, patient acceptability rates rose [28]. Government agencies should actively emphasize the effectiveness and importance of vaccination while publicly addressing concerns about vaccine safety.

A systematic review of access to vaccination by migrants and minorities in many countries highlighted concerns about migration and healthcare systems [29]. Low confidence in COVID-19 vaccines among Black populations, driven by mistrust in government and safety concerns, led to high vaccine hesitancy, according to one systematic review. According to the authors, vaccine hesitancy poses a significant barrier to COVID-19 vaccine uptake in this ethnic minority. For migrants, convenience factors such as language barriers, fear of deportation, and limited physical access have reduced access to COVID-19 vaccines [29].

Al-Metwali B.Z. et al. (2021) remind that public health experts should also attenuate the perceived barriers by providing vaccines and allaying people’s concerns about their storage, effectiveness, and adverse events [10]. Vaccine hesitancy can be addressed by paying careful attention to the application of vaccination programs, the correct and effective use of social media, the transparent and precise management of policy processes, and the provision of evidence-based information about vaccines [20].

A study by Ashok N. et al. (2021) demonstrated the need for additional efforts to address specific concerns and stop the dissemination of false information about vaccines online, particularly among junior-level HCWs and Black, Asian, and minority ethnic (BAME) populations [30]. With effective vaccine promotion campaigns, policymakers should increase public awareness and secure timely and affordable vaccines for the general population [31].

Building trust, reducing physical barriers, and improving communication and transparency about vaccine development through HCWs and religious and community leaders can improve access and facilitate the uptake of COVID-19 vaccines among ethnic minorities and migrant communities [24,30]. The best and most successful policy choices about the pandemic should be shared by countries for COVID-19 and other diseases with similar characteristics.

## 4. Discussion

We reviewed the main reasons for COVID-19 vaccination acceptance and refusal in the HCWs and the general population, providing useful elements for advancing public policies to improve vaccination performance. Reasons for refusal to vaccinate were similar between the HCWs and the general population.

However, there were few multicenter studies, and we excluded commentaries and opinion articles. Most of the study designs were cross-sectional and qualitative. Distrust and refusal by physicians and nurses to be vaccinated are problematic since the HCW is a model of behavior for the general population.

According to a recent systematic review and meta-analysis [31], reminders via mobile phones on interventions to improve coverage and timeliness of routine childhood vaccination enjoyed wide acceptance, thus providing a technological approach that can be used to increase COVID-19 vaccination in the general population.

A survey of attitudes towards the COVID-19 vaccine among Austrian citizens revealed that doctors have a significant impact on people’s decision-making and potential vaccine uptake, especially among older people, due to more frequent consultations for other health problems [32,33]. According to the authors, raising awareness and training doctors, nurses, and pharmacists to conduct vaccination campaigns in close collaboration and synergy with national health authorities can help increase coverage in both urban and rural areas.

Training for physicians, nurses, and pharmacists to conduct vaccination campaigns can help increase coverage in both urban and rural areas. The main driver for healthcare workers to convince the population to be vaccinated is their own belief in the vaccine and their willingness to be vaccinated themselves.

### Limitations of the Study

A significant research drawback of the study is that only 2021 and 2022 were covered by the systemic review, which indicates a dearth of research. Despite broadening the search parameters and language, no papers on vaccination during the pandemic in low- and middle-income countries could be found.

“Arabian countries”, “European countries”, and “other countries” were included in only one global research study [6]. An additional limitation was that only four of the studies dealt with concerns particular to gender [6,9,19,24]. There were no studies that included enough details or proposals to compare groups.

## 5. Conclusions

The major reasons for unwillingness to vaccinate among people during the pandemic were a lack of understanding about the disease, skepticism about the vaccine’s efficacy, and a fear of adverse reactions and serious complications.

The short time it took to develop the vaccine, as well as the lack of transparency concerning pandemic data in some countries, were the main reasons for professionals’ skepticism. On the other hand, the following factors increased professionals’ commitment to vaccination: fear of acquiring the disease, fear of contaminating the family, scientific understanding about the need for immunization, routine treatment of severely ill patients, and an increasing number of deaths.

National vaccination policies for health professionals and the general public included education to prevent false information, organization of the vaccination process with a defined flow from vaccine delivery, packaging, and vaccination, designation of pharmaceutical professionals for vaccination, vaccinator training, and, in some countries, compulsory vaccination.

Low- and middle-income countries faced a vaccination shortage [33]. COVID-19 was a global public health issue, and gaps in vaccination coverage contributed to policy failures and had an impact on the organization of healthcare systems. Poorly organized government mass vaccination strategies have fostered public mistrust in the vaccine’s effectiveness and the need to defeat the pandemic.

Governments and healthcare officials will require guidance in developing and implementing vaccine coverage policies, communicating clearly with the public to control outbreaks involving new strains, and ensuring that vaccination reaches the entire world’s population.

Future studies could help with decision-making on this sensitive subject even more.

## Figures and Tables

**Figure 1 ijerph-21-01134-f001:**
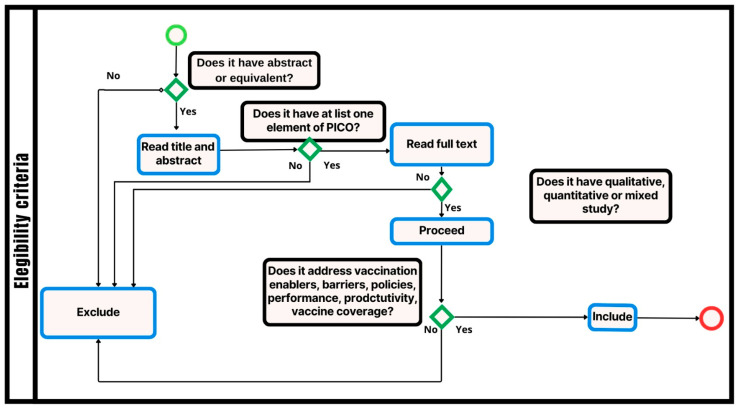
Flowchart of eligibility criteria.

**Figure 2 ijerph-21-01134-f002:**
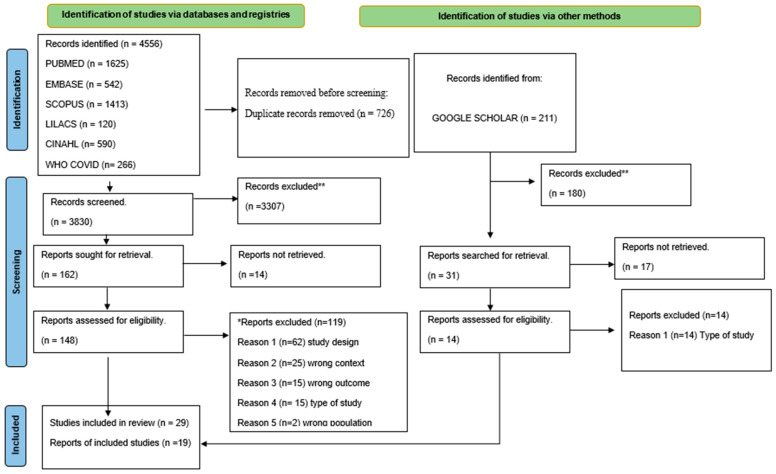
PRISMA flowchart. *: Databases excluded, **: Google Scholar excluded.

**Table 1 ijerph-21-01134-t001:** Terms used to construct the PubMed search strategy.

Question 1				
Keywords	COVID-19	Healthcare Workers	Vaccination	Barriers, Facilitators
Mesh Terms	COVID-19, SARS-CoV-2	Health Personnel, Health Workforce, Caregivers, Health Facilities	COVID-19 Vaccines, Vaccination, Vaccination Coverage	Vaccination Refusal
Entry Terms	Severe Acute Respiratory Syndrome Coronavirus 2, Coronavirus Disease 2019, 2019 Novel Coronavirus, 2019 New Coronavirus, Wuhan Coronavirus, 2019-nCoV, HCoV-19, nCoV-2019, Novel Coronavirus 2019-nCoV, COVID-19 Vaccine, COVID-19 Vaccination	Workforce, Health Manpower, Health Care Provider, Healthcare Provider, Health Care Worker, Healthcare Worker, Health Care Professional, Healthcare Professional	Immunization, Vaccine, Antivaccine, Anti-vaccine, Anti-vaccination, Coverage, OR Infection	Acceptance, Hesitanc, Refusal, Acceptability, Knowledge, Attitude, Practice, Intention, Barrier, Enabler, Determinant
Question 2				
Keywords	COVID-19	Immunization Programs	Barriers and Mechanisms	
Mesh Terms	COVID-19, SARS-CoV-2	Immunization Programs	Vaccination Refusal	
Entry Terms	Severe Acute Respiratory Syndrome Coronavirus 2, Coronavirus Disease 2019, 2019 Novel Coronavirus, 2019 New Coronavirus, Wuhan Coronavirus, 2019-nCoV, HCoV-19, nCoV-2019, Novel Coronavirus 2019-nCoV, COVID-19 Vaccine, COVID-19 Vaccination	Mass Vaccination, Mass Vaccination, Mass Immunization, Vaccination Programm, Vaccine Programm, Vaccination Team, Vaccine Team, Vaccination Campaign, Vaccine Campaign, Vaccination, Anti-vaccine, Anti-vaccination	Acceptance, Hesitanc, Refusal, Acceptability, Barrier, Enabler, Challenge, Facilitator, Performance, Improvement, Optimizer, Strateg, Mechanism, Tool, Productivit, Success	
Question 3				
Keywords	COVID-19	Healthcare Workers	Requirements	
Mesh Terms	COVID-19, SARS-CoV-2	Health Personnel, Health Workforce, Caregivers, Health Facilities	______	
Entry Terms	Severe Acute Respiratory Syndrome Coronavirus 2, Coronavirus Disease 2019, 2019 Novel Coronavirus, 2019 New Coronavirus, Wuhan Coronavirus, 2019-nCoV, HCoV-19, nCoV-2019, Novel Coronavirus 2019-nCoV, COVID-19 Vaccine, COVID-19 Vaccination	Health Workforce, Workforce, Health Manpower, Health Personnel, Health Care Provider, Healthcare Provider, Health Care Worker, Healthcare Worker, Health Care Professional, Healthcare Professional, Caregiver	Requirement, Qualification, Precondition, Requisite, Prerequisite, Abilit, Skill	
Question 4				
Keywords	COVID-19	Healthcare Workers	Vaccination	Policies
Mesh Terms	COVID-19, SARS-CoV-2	Health Personnel, Health Workforce, Caregivers, Health Facilities	COVID-19 Vaccines, Vaccination, Vaccination Coverage	Mandatory Programs, Policy
Entry Terms	Severe Acute Respiratory Syndrome Coronavirus 2, Coronavirus Disease 2019, 2019 Novel Coronavirus, 2019 New Coronavirus, Wuhan Coronavirus, 2019-nCoV, HCoV-19, nCoV-2019, Novel Coronavirus 2019-nCoV, COVID-19 Vaccine, COVID-19 Vaccination	Workforce, Health Manpower, Health Care Provider, Healthcare Provider, Health Care Worker, Healthcare Worker, Health Care Professional, Healthcare Professional	sars-cov-2 vaccine, COVID-19 Vaccination, Immunization, Vaccine	Political, Regulation, Prioritization, Prioritize Prioritise, Mandat, Law, Laws, Right, Rule, Obligatoriness, Regulatory

## Data Availability

The database of the literature review is available upon request from the authors. PROSPERO registration number: CRD42022325111.

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
