# Peer review of "COVID-19 Vaccination Acceptance and Hesitancy in Healthcare Workers and the General Population: A Systematic Review and Policy Recommendations"

_ijerph, 2024, doi:10.3390/ijerph21091134_

Round 1
Reviewer 1 Report
Comments and Suggestions for Authors
Overall, the manuscript quality is excellent and the contents are interesting for readers. I will recommend publishing the manuscript following some minor general comments.
1 . The introduction is good however, I would advice the authors to refer the study aim and objectives so the reader can easily follow.
2. The figure's quality can be improved as some text can not be seen.
3. The study conclusion section is too long to my understanding, please brief it.
Comments on the Quality of English Language
Minor editing of the English language is needed.
Author Response
Comment on the language: Minor editing of the English language is needed.
Response: A translator revised the text.
Overall comment: Overall, the manuscript quality is excellent, and the contents are interesting for readers. I will recommend publishing the manuscript following some minor general comments.
Response: Many thanks for your remarks.
Comment 1 . The introduction is good however, I would advise the authors to refer the study aim and objectives so the reader can easily follow.
Response 2.The objectives are laid out in lines 104-106 of the introduction
The study aims to examine factors that influence HCWs` and general population’s adherence to vaccination and elucidate policies and strategies to cover health personnel and the population at large with COVID-19 vaccination.
Comment 2. The figure's quality can be improved as some text can not be seen.
Response 2. The resolution of the two figures was improved.
Comment 3. The study conclusion section is too long to my understanding, please brief it.
Response 3. The conclusion has been reworded to be more concise
Reviewer 2 Report
Comments and Suggestions for Authors
Summary
In this manuscript, the authors investigate the factors that influence health and care workers’ (HCW) adherence to COVID-19 vaccination and national policies to vaccinate the HCW and other target groups using systematic review method. The authors included nineteen papers in this review. They found the evidence is more related to vaccine hesitancy rather than acceptance, the concerns are about safety since the vaccine is developed rapidly and is not sufficient tested. Specific comments are listed below:
1. All figures are not readable.
2. No description of figures in context.
3. Does the age affect the acceptance of vaccination? What’s the percentage of HCW and general populations accept vaccine?
Author Response
Comment 1: All figures are not readable.
Response 2: The presentation of findings has been adjusted.
Comment 2: No description of figures in context.
Response 2: The text includes descriptions of the figures.
Comment 3: Does the age affect the acceptance of vaccination? What’s the percentage of HCW and general populations accept vaccine?
Response 3: Although the comment is relevant, these variables weren't found in the literature reviewed, hence it is impossible to answer to the
Reviewer 3 Report
Comments and Suggestions for Authors
Thank you so much for submitting your paper in our journal. Please go through the comments to improve your manuscript.
Introduction:
- The introduction does not provide sufficient background on the specific challenges and successes of COVID-19 vaccination programs in different regions, particularly in low- and middle-income countries (LMICs).
- The introduction makes broad, generalized statements about vaccine hesitancy without citing specific studies or data to support these claims, which weakens the argument.
Methods:
- The methods section lacks a detailed explanation of the search strategy used, including specific keywords, databases searched, and the criteria for selecting studies.
- The inclusion and exclusion criteria for the studies are not justified or explained thoroughly, making it difficult to assess the validity of the selection process.
- The process of data extraction is not described in detail, including how disagreements between reviewers were resolved and how data was managed.
Results:
- The results section reports data inconsistently, with some studies providing detailed findings while others are summarized briefly, making it hard to compare results across studies.
Discussion:
- The discussion lacks a critical analysis of the findings, including potential limitations and biases of the included studies, which is crucial for a systematic review.
- The discussion does not adequately address the key issues of vaccine hesitancy among healthcare workers and the general population, nor does it suggest concrete solutions.
11. The discussion section lacks sufficient references to support the claims made. Including more references would strengthen the discussion by providing evidence from additional studies that corroborate the findings. This is particularly important for claims about the effectiveness of policy interventions and public health strategies. Please 3-4 references, the discussion section will be more robust and credible, as it will be backed by a wider range of studies and evidence. This approach not only strengthens the claims made but also situates the findings within the broader landscape of existing research, making the conclusions more convincing and well-founded. You can use the following reference if applicable:
The Long and Winding Road: Uptake, Acceptability, and Potential Influencing Factors of COVID-19 Vaccination in Austria. https://doi.org/10.3390/vaccines9070790
Conclusion:
- The conclusion offers weak and generic recommendations that do not provide clear, actionable steps for policymakers to address vaccine hesitancy and improve vaccination rates.
Minor editing
Author Response
Comment on the language: Minor editing.
Response: A translator revised the text.
Introduction:
Comment 1: The introduction does not provide sufficient background on the specific challenges and successes of COVID-19 vaccination programs in different regions, particularly in low- and middle-income countries (LMICs).
Response: Lines 71–77 of the introduction contain a paragraph with information regarding low-income countries.
“High-income countries secured their vaccine stocks in advance by pre-purchasing millions of doses of vaccines even before the clinical trials were finalised. Asian countries such as Japan and Australia have also followed this trend and secured their vaccine doses. However, for low-income countries, such as those in Africa, the trend of purchasing and guaranteeing early access has not occurred. As a result, more than 3 billion doses of the COVID-19 vaccination have been administered worldwide by July 2021, with only 0.9% of people in low-income countries having received at least one dose. (3).”
Comment 2: The introduction makes broad, generalized statements about vaccine hesitancy without citing specific studies or data to support these claims, which weakens the argument.
Response: Lines 84–97, of the introduction, refer to four studies (references 6, 7, 8, and 9) that address the topic and suggest that the reasons for not vaccination are related to variables such as "efficacy, safety, and the accelerated and unprecedented process of vaccine production".
Methods:
Comment 3: The methods section lacks a detailed explanation of the search strategy used, including specific keywords, databases searched, and the criteria for selecting studies.
Response:
The databases are described in lines 162 to 167.
“The following databases were searched for this review: MEDLINE-PubMed, Em-base, Scopus, Latin American and Caribbean Health Sciences Literature, and Web of Science. The World Health Organization Database and Google Scholar were adopted for the grey literature”.
The search terms used are described in lines. 167 a 169.
“The controlled vocabularies for the health area DeCs (Descriptors in Health Sciences), MeSH (Medical Subject Headings), and Emtree (Embase Subject Headings) were consulted to retrieve findings. The search terms used were a combination of official descriptors and free terms, extracted from the controlled vocabularies Mesh, DeCS, and Emtree, which enabled the construction of search strategies for each database adopted”.
Annexe 3 presents all of the search strategies by database used.
Comment 4: The inclusion and exclusion criteria for the studies are not justified or explained thoroughly, making it difficult to assess the validity of the selection process.
Response: The eligibility criteria are included in the text on lines 151–157, as well as in Annex 1.
“The following were considered eligible for research: studies in which the target population is or includes HCWs; studies that address vaccination against COVID-19; studies that address the requirements of health professionals for the population to accept the vaccine; studies that describe policies and strategies for vaccine acceptance by the population; studies that describe approaches to improving the quality and productivity of vaccination teams and studies that describe facilitators and barriers to COVID-19 vaccination among healthcare Workers”.
Comment 5: The process of data extraction is not described in detail, including how disagreements between reviewers were resolved and how data was managed.
Response: The differences were resolved as reported on lines 181 to 182 and lines 188 to 191.
Differences were discussed with a third researcher until a consensus was reached.
In case of disagreement, a discussion between the two reviewers was done, and a third reviewer was consulted. Kappa statistics were computed to assess inter-reviewer agreement and the quality of this process. Kappa inter-reviewer agreement was 9.4 (14).
Results:
Comment 6: The results section reports data inconsistently, with some studies providing detailed findings while others are summarized briefly, making it hard to compare results across studies.
Response: The results of the review were extracted from the included studies according to the available results. We believe that the reviewer's perception is fair, but the inconsistency is possibly due to the very heterogeneity of the studies on the subject. We tried to report the main findings of the included studies, but comparing the results per se would depend on standardised outcomes, which was not the purpose of the systematic review.
There are not comments 7 and 8
Discussion:
Comment 9: The discussion lacks a critical analysis of the findings, including potential limitations and biases of the included studies, which is crucial for a systematic review.
Response: We reviewed the main reasons for COVID-19 vaccination acceptance and refusal in the HCW and the general population, providing useful elements for advancing public policies to improve vaccination performance. However, there were few multicenter studies and we excluded commentaries and opinion articles. Most of the study designs were cross-sectional and qualitative. The documents and articles published from 2020 to 2022 displayed various problems with reliability. Distrust and refusal by physicians and nurses to be vaccinated is problematic, since the HCW is a model of behavior for the general population. More research is needed to report on campaigns and strategies to improve the productivity of vaccination teams.
Comment 10: The discussion does not adequately address the key issues of vaccine hesitancy among healthcare workers and the general population, nor does it suggest concrete solutions.
Response: The major reasons for unwillingness to vaccinate among the people during the pandemic were a lack of understanding about the disease, skepticism about the vac-cine's efficacy, and a fear of adverse reactions and serious complications. The short time it took to develop the vaccine, as well as the lack of transparency concerning pandemic data in some countries, were the main reasons for professionals' skepticism.
Comment 11: The discussion section lacks sufficient references to support the claims made. Including more references would strengthen the discussion by providing evidence from additional studies that corroborate the findings. This is particularly important for claims about the effectiveness of policy interventions and public health strategies. Please 3-4 references, the discussion section will be more robust and credible, as it will be backed by a wider range of studies and evidence. This approach not only strengthens the claims made but also situates the findings within the broader landscape of existing research, making the conclusions more convincing and well-founded. You can use the following reference if applicable:
The Long and Winding Road: Uptake, Acceptability, and Potential Influencing Factors of COVID-19 Vaccination in Austria. https://doi.org/10.3390/vaccines9070790
Response: As suggested, 2 references have been included (33 and 34)
- King I, Heidler P, Marzo RR. The Long and Winding Road: Uptake, Acceptability, and Potential Influencing Factors of COVID-19 Vaccination in Austria. Vaccines (Basel). 2021 Jul 15;9(7):790. doi: 10.3390/vaccines9070790. PMID: 34358206; PMCID: PMC8310144.
- Noushad M, Nassani MZ, Al-Awar MS, et al. COVID-19 Vaccine Hesitancy Associated With Vaccine Inequity Among Healthcare Workers in a Low-Income Fragile Nation. Front Public Health. 2022;10:914943. Published 2022 Jul 11. doi:10.3389/fpubh.2022.914943
Conclusion:
Comment 12: The conclusion offers weak and generic recommendations that do not provide clear, actionable steps for policymakers to address vaccine hesitancy and improve vaccination rates.
Response: The conclusions were completely rewritten to better present both the main reasons of hesitancy to vaccinate among HCWs and the general population, as well as the national strategies and policies in place at the time for vaccinating health professionals and the general population, as reported in the systematic review studies.
Reviewer 4 Report
Comments and Suggestions for Authors
1. There is no connectivity in sections 3.4 and sections 3.5. Please rewrite both the sections.
2. Discussion is laking the continuity
3. Conclusion must be rephrased and written very precisely.
4.Current policy recommendations must be updated in manuscript.
Comments on the Quality of English LanguageFine.
Author Response
Comment 1. There is no connectivity in sections 3.4 and sections 3.5. Please rewrite both the sections.
Response: Section 3.4 and 3.5 have been reworded
Comment 2. Discussion is laking the continuity
Response: The conclusions have been rewritten
Comment 3. Conclusion must be rephrased and written very precisely.
Response: The conclusion has been revised to include both the primary causes of hesitancy to vaccinate among HCWs and the general population, as well as the national strategies and policies in place at the time for vaccinating health professionals and the general population, as reported in the systematic review studies.
Comment 4. Current policy recommendations must be updated in manuscript.
Response: The study's objective was to reveal, by analysing the selected documents, the policies and/or strategies adopted by national governments at that time.
Reviewer 5 Report
Comments and Suggestions for Authors
The article is well structured, however some clarifications are needed. First, was a PRISM made? Because if this were the case it would be necessary to cite it and modify figure 2 like that of PRISMA, here is where to take the reference https://www.prisma-statement.org/prisma-2020-flow-diagram. The resolution of figure 1 and 2 is too low and should be improved. Furthermore, although the study was included in PROSPERO, it does not introduce anything new and, in my opinion, lacks originality. Therefore either a diagram to improve vaccination hesitancy among the general population should be proposed or it should be rejected.
Author Response
Comment: The article is well structured, however some clarifications are needed.
Response: The presentation of the results has been revised.
Comment: First, was a PRISM made? Because if this were the case it would be necessary to cite it and modify figure 2 like that of PRISMA, here is where to take the reference https://www.prisma-statement.org/prisma-2020-flow-diagram.
Response: Line 119 indicated that the study planning was directed by the PRISMA 2020 Declaration, and the illustration was changed to depict the PRISMA flowchart.
Comment: The resolution of figure 1 and 2 is too low and should be improved.
Response: The resolution of the two figures was improved.
Comment: Furthermore, although the study was included in PROSPERO, it does not introduce anything new and, in my opinion, lacks originality. Therefore either a diagram to improve vaccination hesitancy among the general population should be proposed or it should be rejected.
Response: We appreciate your suggestions; however it was not the goal of the review. Given the existing knowledge, we could consider include the recommendation in a future article.
Round 2
Reviewer 5 Report
Comments and Suggestions for Authors
The article has been sufficiently improved, although I repeat that it is not of great scientific relevance. References should be implemented.
Author Response
Comment: The article has been sufficiently improved, although I repeat that it is not of great scientific relevance. References should be implemented.
Response: We appreciate the reviewer's remarks on the article's sufficient improvement, but we respectfully disagree with his opinion of its scientific importance.
The COVID-19 epidemic has posed one of the most significant public health issues of the twenty-first century. In this setting, vaccination has emerged as one of the most effective strategies for controlling viral spread and mitigating illness effects. However, acceptability and reluctance to vaccinate among health professionals and the general public has been and continues to be a significant scientific and political issue.
First and foremost, health professionals have an important role in promoting vaccination and fostering public trust. Their acceptance and adherence to vaccination can have a direct impact on the population's readiness to follow health recommendations (Verger et al., 2018). Understanding the elements that influence these professionals' views towards vaccination is critical for developing effective vaccination promotion programmes.
Furthermore, vaccine reluctance among the general population remains a substantial impediment to vaccination coverage and herd immunity. According to research, trust, convenience, and complacency are important factors influencing vaccine acceptance (WHO, 2014). Analyzing these elements in the context of a pandemic can yield significant insights for developing more effective policies and actions.
In this regard, a review of the acceptance and hesitancy of COVID-19 vaccination among health professionals and the general population is highly scientifically relevant and could assist to understand the main determinants of vaccination, facilitate informed decision-making, and support communication and community engagement strategies (Lazarus et al., 2021). Such measures are critical to achieving high vaccine coverage and so more effectively controlling the epidemic.
References:
Lazarus, J. V., Ratzan, S. C., Palayew, A., Gostin, L. O., Larson, H. J., Rabin, K., Kimball, S., & El-Mohandes, A. (2021). A global survey of potential acceptance of a COVID-19 vaccine. Nature Medicine, 27(2), 225-228.
Verger, P., Fressard, L., Collange, F., Gautier, A., Pulcini, C., Peretti-Watel, P., ... & Launay, O. (2018). Vaccine hesitancy among general practitioners and its determinants during controversies: a national cross-sectional survey in France. EBioMedicine, 27, 236-242.
World Health Organisation (WHO) (2014). Report of the SAGE working group on vaccine hesitancy. Geneva: WHO.